

PeerJ Hubs
Published on behalf of


International Association for Biological Oceanography
IABO

# Cnidom in Ceriantharia (Cnidaria, Anthozoa): new findings in the composition and micrometric variations of cnidocysts

Agustín Garese[1],[*], Fabiola Goes Correa[2],[*], Fabián H. Acuña[1],[3] and Sérgio Nascimento Stampar[2]

[1] Institute of Marine and Coastal Research (IIMyC), Faculty of Exact and Natural Sciences, National University of Mar del Plata (UNMDP) and The National Scientific and Technical Research Council (CONICET), Mar del Plata, Buenos Aires, Argentina
[2] Faculty of Sciences, Department of Biological Sciences, Laboratory of Evolution and Aquatic Diversity—LEDA, Sao Paulo State University "Júlio de Mesquita Filho" (UNESP), Bauru, Sao Paulo, Brazil
[3] Scientific Station COIBA (COIBA-AIP), Clayton, Panamá
[*] These authors contributed equally to this work.

Corresponding author
Agustín Garese, agustingarese@gmail.com

## ABSTRACT

**Background:** Like all cnidarians, the subclass Ceriantharia (Cnidaria, Anthozoa) is known for producing cnidocysts, which mainly serve for prey immobilization, predator defense, and locomotion.

**Aim:** The present study aimed to understand the variability of the cnidom, *i.e.*, the inventory of all cnidocyst types, in the ceriantharians (tube anemones) *Ceriantheomorphe brasiliensis* (10 individuals) and *Cerianthus* sp. (seven individuals).

**Methods:** In each individual, 30 intact cnidocysts of each identified type were measured in the following parts of tube anemones: marginal tentacles (four from each individual), labial tentacles (four from each individual), column, actinopharynx and metamesenteries. Each of these structures was divided into three levels (high, middle, and low) and the cnidom was analyzed. Statistical descriptive parameters (mean, standard deviation, minimum and maximum) of the sizes of all types of cnidocysts were calculated. The normality of the data for cnidocyst length was assessed using a Shapiro-Wilk test ($\alpha = 0.05$). Based on the acceptance or rejection of the normality, either linear models or generalized linear models were used to evaluate variations in cnidocyst lengths. The normality of the cnidocyst length was tested by Shapiro-Wilk, and due to its rejection, generalized linear mixed models were applied to test the cnidocyst lengths variations.

**Results:** The analysis of *Ceriantheomorphe brasiliensis* revealed 23 categories of cnidocysts, thereby expanding the understanding of its cnidome. The cnidoms of *Ceriantheomorphe brasiliensis* and *Cerianthus* sp. presented intraspecific variations, both qualitatively and in the lengths of cnidocysts. The cnidoms of the two species studied also showed qualitative intra-individual variations between different levels (high, middle, low) within each structure of the tube anemone (tentacles, actinopharynx, column and metamesenteries). Some cnidocyst types, such as atrichs from the column of *C. brasiliensis*, presented a length gradient along the column, from larger lengths at the "low" level to smaller lengths at the "high" level.

**Conclusions:** The cnidom of a tube anemone could be better described if samples are taken at different levels of the structures, as observed in *C. brasiliensis*. In addition, we can conclude that the cnidocyst lengths of both *C. brasiliensis* and *Cerianthus* sp. present intraspecific variation, which is coincident with that observed in actiniarian sea anemones. Moreover, as main conclusion, this work also proved that individuals of tube anemone species could present qualitative intra-structure variations in both the cnidom and cnidocyst lengths. This characteristic appears as an exception in cnidom variations, and has so far not been recorded even in the most studied actiniarian sea anemones. Finally, the intra-structure cnidocyst variations could reveal different functions of the different levels of a particular body part of the organisms.

## INTRODUCTION

The Phylum Cnidaria is known for producing cnidocysts, which are capsules containing thread-like tubes. These intracellular structures are classified into three primary types: nematocysts, ptychocysts and spirocysts. Nematocysts and spirocysts are responsible for directly assisting in the capture of prey, aggression and defense of the individual (*Fautin, 2009*), while ptychocysts are involved in the construction of the tube of ceriantharian anemones (Cnidaria, Anthozoa, Ceriantharia) (*Mariscal, Conklin & Bigger, 1977*; *Stampar et al., 2015*). In all cnidarians, cnidocysts show great diversity of shapes and sizes, which are considered useful to characterize some genera or species (*Weill, 1934*; *Schmidt, 1969*; *Fautin, 2009*; *Pica & Puce, 2017*).

In terms of morphology, the most diverse cnidocysts are nematocysts. In the early 1930s, *Weill (1930, 1934)* made a thorough classification of nematocysts, recognizing about 30 different morphologies. This classification has been the most used and debated since its publication. Subsequently, other authors proposed modifications to make Weill's classification clearer by adding the morphology of newly discovered nematocysts (*Carlgren, 1940*; *Cutress, 1955*; *Den Hartog, 1977*; *England, 1991*; *Östman, 2000*) or proposed a new classification for nematocysts of Anthozoa (*Schmidt, 1969, 1972, 1974*). The identification of nematocysts was based on characters as: the shaft form in undischarged cnidocysts, the filament length in relation to the capsule length, the disposition of spines, and the presence or absence of a terminal tubule (*England, 1991*). The diversity of nomenclatures led *Fautin (1988)* to suggest that, in every publication, nematocyst classification should be illustrated to improve the communication between specialists following different nomenclatures.

*Mariscal (1974)* proposed that cnidocysts are involved in both offensive and defensive functions, whereas *Ewer & Fox (1947)* suggested that, from a functional point of view, cnidocysts can be divided in three types: penetrants, volvents, and glutinants. Penetrants are those that present a shaft, such as those defined as *p* or *b* mastigophores or

amastigophores (*Östman, 2000*), whose main function is related to prey capture. Volvents are those that present spines but not a defined shaft, such as those called atrichs (with very few spines according to *Cutress (1955)*), basitrichs and holotrichs (*Östman, 2000*), and whose functions are linked to defense and/or aggression. Glutinants are involved in the adhesion to the substrate, locomotion or prey capture, being the iconic type spirocysts. However, beyond their morphology, the anatomical location of cnidocysts may indicate much about their function (*Shick, 1991*).

Knowledge about cnidocysts has advanced in different aspects since their discovery, including their usefulness or not in taxonomy. As some types of cnidocysts are found only in specific groups, *Stephenson (1929)* stated that species and/or genera of sea anemones of the subclass Actiniaria (Cnidaria: Anthozoa) can be differentiated based on the characteristics of their cnidocysts. In fact, several authors have considered that, in general, the description of cnidocysts and their respective measurements is an ally in the taxonomy of Anthozoa (*Carlgren, 1940*; *Cutress, 1955*; *Shick, 1991*), and *Carlgren (1940)* pointed out that no species description is complete unless it includes a description of the cnidom (*i.e.*, the inventory of all cnidocyst types). However, a study conducted by *Williams (1996)* showed that the cnidom can vary within the same species, thus questioning its usefulness for taxonomic purposes. It should be considered that, in Actiniaria, for example, the size and type of cnidocysts may vary according both to the environmental conditions to which the animal is subject and to the size of the individual, and that distinct cnidocysts can be present in some structures (*Francis, 2004*; *Acuña, Excoffon & Ricci, 2007*; *Fautin, 2009*). Currently, studies on the cnidom already cover statistical methods to test the intraspecific variations of the sizes of these structures, as presented in *Garese, Carrizo & Acuña (2016)*. These studies have shown that, at least in actiniarian sea anemones, the intraspecific variation in the cnidocyst size is the rule rather than the exception (*Garese, Carrizo & Acuña, 2016*), and that, consequently, the taxonomic value of these data is doubtful. However, quantitative analyses to distinguish closely related species or morphotypes of the same species suggest that the differences in the sizes of their cnidocysts are statistically significantly (*González-Muñoz et al., 2017*; *Maggioni et al., 2021*). On the other hand, other works have found no statistical support to distinguish specimens based on the differences between the sizes of their cnidocysts (*González-Muñoz et al., 2018*).

Regarding intra-structure variations, *Williams (1996*, *1998*, *2000)* analyzed the cnidocysts of five families of sea anemones and found that different samples of the same structure (mentioned as "tissue") showed differences between the lengths of cnidocysts. Consequently, the mentioned author pointed out that, when sampling cnidocysts by using the classical approach followed to study the cnidom in sea anemones, the section of each structure should be clearly identified. The classical approach of the study of cnidocysts implies sampling cnidocysts in all the structures present in the species, taking portions of structures at a particular level (*Williams, 1996*), such as the middle region of the column, the tips of tentacles, or the middle level of actinopharynx, *etc*. This methodology supposed certain uniformity of the presence of certain cnidocyst types along a structure. Later, by studying a single specimen of *Actinodendron arboretum*, *Ardelean & Fautin (2004)* reinforced the idea of *Williams (1996)* and suggested that the sampling site of a structure

(mentioned as "tissue") could be an important variable to determine cnidocyst length. These authors also stated that unifying a site of sampling is useful and necessary to make comparisons between individuals or species (*Ardelean & Fautin, 2004*). However, considering the studies that originated it (*Williams, 1996*; *Ardelean & Fautin, 2004*), the above-mentioned classical approach turns out to be quite contradictory because if the cnidocyst length could vary depending on the site of sampling in a particular structure, determining a particular level of a structure would not represent the variability of sizes of cnidocysts present in the other levels of a structure. In fact, to our knowledge, there are no works that have analyzed the qualitative variation of the cnidom between levels in a structure.

Although there is considerable knowledge in Anthozoa, in general, there is no information about the variations of the cnidom in the subclass Ceriantharia. The cnidom of some species of this subclass, such as *Arachnanthus australiae* (Carlgren, 1937), *Pachycerianthus curacaoensis* (*Den Hartog, 1977*), *Isarachnanthus nocturnus* (*Den Hartog, 1977*) and *Botruanthus mexicanus* (*Stampar, González-Muñoz & Morandini, 2017*), has been described and compared to that of sea anemones by a few authors. However, there are no studies highlighting the variability and micrometrics of the cnidom in detail. Although limited, the study of the cnidom of Ceriantharia helps as one of the main resources of identification due to the highly difficult collection of these animals (*Spier, Stampar & Prantoni, 2012*). In consequence, the present study aimed to test the variability of the cnidom in *Ceriantheomorphe brasiliensis* (Mello-Leitão, 1919) and *Cerianthus* sp., as study cases in Ceriantharia, including an analysis of the variation in the length of cnidocysts with a novel approach of sampling at three different levels of each body part of the organisms.

# MATERIALS AND METHODS

## Species and number of specimens studied

All specimens were collected manually by SCUBA diving and preserved in 4% formaldehyde. The cnidoms of ten specimens of *Ceriantheomorphe brasiliensis* and seven specimens of *Cerianthus* sp. were analyzed. However, for *Cerianthus* sp., the number of individuals used to study the cnidocysts was variable in the different structures of the species (Table S1). Since Ceriantharians are very difficult to find and collect, the availability of specimens is often very low. Moreover, some specimens can suffer some kind of deterioration according to the time spent in the collection itself or due to their uses in different previous researches, such as those for taxonomic identification. In consequence, the cnidom analysis in the structures of *Cerianthus* sp. was carried out in the following number of specimens for each structure: six for the actinopharynx, seven for the column, three for the metamesenteries, seven for the labial tentacles, and four for the marginal tentacles.

## General cnidom analysis

Whenever possible (*i.e.*, before using up the tissue available), 30 intact capsules of each cnidocyst type identified in each structure of each specimen were measured by "*squash*" preparations. For this purpose, a Nikon Eclipse E200 microscope at 1000× and the Motic

Images Plus 2.0 software were used. The cnidom of the following body parts of the tube anemones were analyzed: marginal tentacles (four tentacles from each specimen), labial tentacles (four tentacles from each specimen), column, actinopharynx and metamesenteries. Each body party was sampled at three independent levels: low, middle and high (Fig. 1). Herein, "level" refers to the relative position of sampling respect to the aboral end of the organisms and "structure" refers to the body parts mentioned above. The nomenclature of cnidocyst types was based on *Mariscal (1974)*, *Mariscal, Conklin & Bigger (1977)* and *Östman (2000)*. A total of 25,317 cnidocysts were measured. The cnidom of each structure was described. Statistical descriptive parameters of cnidocyst sizes (mean, standard deviation, minimum and maximum) were calculated in all types of cnidocyst found. Then, the length of all cnidocyst types was compared between individuals without discriminating between levels in this case. Only cnidocyst length data were used for comparisons since the width of cnidocyst data are very little variable (*Garese, Carrizo & Acuña, 2016*). The normality of the distribution of the cnidocyst length was tested by the Shapiro-Wilk test ($\alpha = 0.05$) on the residuals of a linear model with normal distribution. In cases where normality was accepted, analysis of variance (ANOVA) was used to test differences between individuals. In data sets in which normality was not accepted, a generalized linear model (GLM) was fitted with gamma distribution for errors and inverse as link function (following *Garese, Carrizo & Acuña, 2016*).

The model used was:

$$g(\text{cnidocyst length}) = \beta_0 + \beta_1(\text{individual}) + \varepsilon$$

where "cnidocyst length" is a dependent variable '$y$', "individual" is an independent variable '$x$', $\beta_0$ is the '$y$' intercept parameter, $\beta_1$ is the parameter estimated for '$x$', '$\varepsilon$' is the statistical error and '$g()$' is the link function.

Then, a T-test ($\alpha = 0.05$) for the $\beta_1$ parameters was used to evaluate differences in the cnidocyst sizes between individuals.

## Intra-structure composition of the cnidom

To determine the intra-structure composition of the cnidom, each structure was separated and analyzed at different levels: low, middle, and high. The number of individuals that presented each type of cnidocyst at each level was recorded, and the percentage of occurrence of all cnidocyst types was calculated for each level in each structure. Then, these percentages were used to produce radar charts by using the R package ggplot2 (*Wickham, 2016*).

A statistical comparison of cnidocyst length was made between levels. This analysis was carried out only in cases where the cnidocyst type was present at the three levels of a structure of all specimens studied or at least in 90% of them. A linear model (LM), linear mixed model (LMM), generalized linear model (GLM) or a generalized linear mixed model (GLMM) was fitted after testing for the normality of the residuals. The general model form was:

$$\text{cnidocyst length} \sim \beta_0 + \beta_1\text{level} + (1 \mid \text{Individual}) + \varepsilon,$$
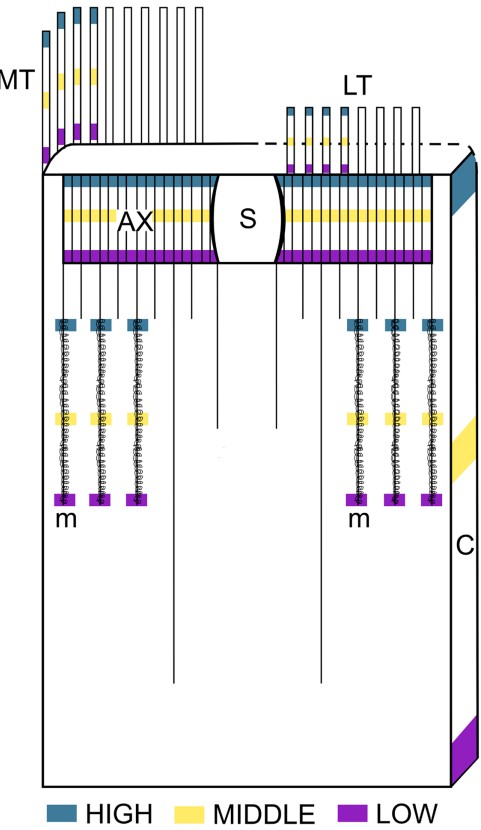

**Figure 1 Schematic representation of the intra-structure samplig of cnidocysts implemented in this study.** Ax, Actinopharynx; C, Column; LT, Labial tentacles; MT, Marginal tentacles; m, Metamesenteries; S, Siphonoglyph.

where the "level" variable was considered as fixed effect and the "individual" variable was considered as random effect (applies only to mixed models) because several measures were taken in each individual. In cases where normality was rejected, a GLM or GLMM with Gamma distribution for errors and identity link function was fitted (following *Garese, Carrizo & Acuña, 2016*). Then, confident intervals of cnidocyst lengths for each level were calculated from the fitted model, and compared.

Also, Kernel density plots (*Sheather, 2004*) were produced to graphically explore the variations of cnidocyst sizes between levels. The density plots were obtained for the cnidocysts that were present at the three levels in more than 70% of the individuals sampled.

All statistical analyses were performed with the R program (*R Core Team, 2022*). The models were produced with the R package 'lme4' (*Bates et al., 2015*). All graphics were made using the ggplot2 R package (*Wickham, 2016*).

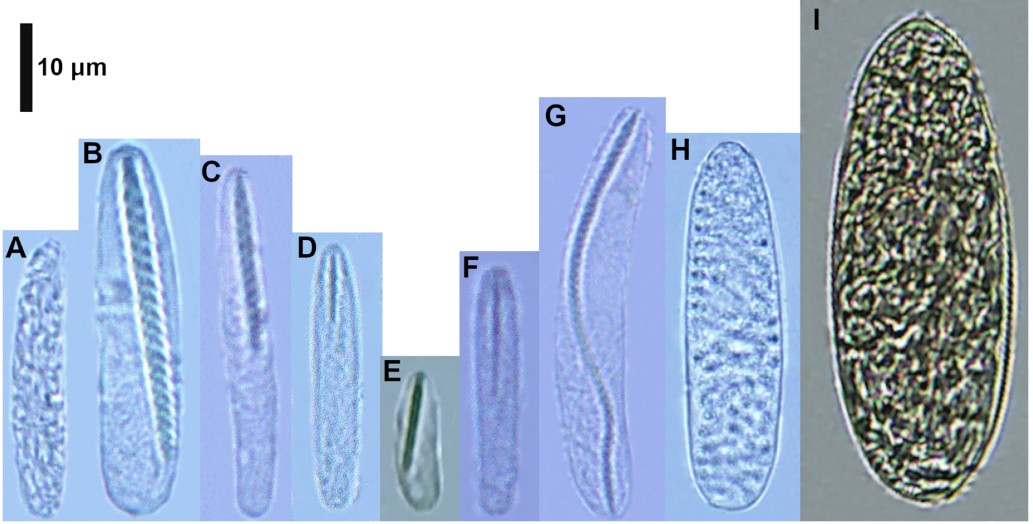

**Figure 2 Cnidocysts of *Ceriantheomorphe brasiliensis*.** (A) Atrich. (B) Microbasic b-mastigophore I. (C) Microbasic b-mastigophore II. (D) Microbasic b-mastigophore III. (E) Microbasic b-mastigophore IV. (F) Microbasic b-mastigophore V. (G) Microbasic b-mastigophore VI. (H) Holotrich. (I) Ptychocyst.

## RESULTS

### *Ceriantheomorphe brasiliensis*
### General cnidom analysis

The cnidom of *Ceriantheomorphe brasiliensis* presented a total of nine cnidocyst types (Fig. 2). The number, length and width of each of the nine cnidocyst types found are reported in Table 1. From all data sets of cnidocyst lengths obtained, only four fitted to a normal distribution (all different cnidocyst types in the structures sampled). Three out of a total of 20 data sets were not analyzed due to a low "N" (see Table 1). These data sets corresponded to microbasic b-mastigophores I (b I) and III (b III) from the actinopharynx, microbasic b-mastigophores V (b V) from the column, and atrichs from the marginal tentacles. The ANOVA produced to test the variation of cnidocyst sizes between individuals indicated significant differences in all cnidocysts analyzed (Actinopharynx: b I (F = 39.07; $P < 0.001$), b II (F = 61.6; $P < 0.001$); Column: b V (F = 109.8; $P < 0.001$); Marginal tentacle: atrichs (F = 135.3; $P < 0.001$)).

For the remaining 16 data-sets, GLMs were applied to test differences in cnidocyst lengths between individuals, showing significant differences (Table S2). In general, all the cnidocysts analyzed, independently of the type and structure analyzed, showed differences in their lengths between individuals.

### Intra-structure qualitative variations of cnidocyst types

Regarding the qualitative composition of the cnidom of *C. brasiliensis*, the different levels of each of the structures analyzed showed different patterns of variations.

**Table 1 Cnidom composition of *Ceriantheomorphe brasiliensis*.** Length and width units: μm; *n* = number of specimens that present cnidocysts/total specimens; *N* = total number of cnidocysts sized. Underlined *P*-values significant at α = 0.05; # Shapiro Test was not applied due to low *N*.

| Structure/Cnidocyst type | Length (mean ± SD) | Length range (min–max) | Width (mean ± SD) | Width range (min–max) | *n* | *N* | *P*-value |
|---|---|---|---|---|---|---|---|
| **Actinopharynx** | | | | | | | |
| Atrich | 38.74 ± 5.80 | 27.09–56.28 | 6.98 ± 1.46 | 3.96–11.79 | 4/10 | 570 | <0.001 |
| Microbasic b-mastigophore I | 47.99 ± 6.94 | 31.97–67.98 | 8.45 ± 2.64 | 3.12–14.52 | 6/10 | 240 | 0.054 |
| Microbasic b-mastigophore II | 41.93 ± 2.80 | 37.53–47.29 | 6.18 ± 0.66 | 4.45–7.16 | 1/10 | 30 | # |
| Microbasic b-mastigophore III | 30.41 ± 5.29 | 19.38–45.92 | 3.67 ± 0.75 | 1.86–5.59 | 9/10 | 389 | 0.854 |
| **Column** | | | | | | | |
| Atrich | 49.62 ± 7.45 | 28.86–75.74 | 12.26 ± 2.63 | 4.97–22.44 | 10/10 | 895 | <0.001 |
| Microbasic b-mastigophore I | 34.02 ± 4.03 | 26.36–46.89 | 6.76 ± 1.05 | 4.13–10.07 | 4/10 | 120 | <0.001 |
| Microbasic b-mastigophore V | 29.04 ± 2.98 | 23.52–36.02 | 3.76 ± 0.65 | 2.29–5.69 | 3/10 | 179 | 0.815 |
| Microbasic b-mastigophore VI | 53.15 ± 2.71 | 46.55–58.46 | 5.82 ± 0.86 | 4.37–8.02 | 1/10 | 30 | # |
| Holotrich | 46.63 ± 4.81 | 33.06–55.04 | 10.33 ± 1.91 | 6.23–16.34 | 1/10 | 30 | # |
| Ptychocyst | 71.97 ± 8.02 | 53.30–92.93 | 28.16 ± 4.36 | 16.08–44.12 | 8/10 | 330 | 0.006 |
| **Metamesenteries** | | | | | | | |
| Microbasic b-mastigophore I | 57.12 ± 7.02 | 32.02–73.54 | 12.45 ± 2.35 | 6.07–18.54 | 4/10 | 360 | <0.001 |
| Microbasic b-mastigophore IV | 19.67 ± 3.44 | 13.55–30.21 | 4.77 ± 1.12 | 3.04–8.28 | 4/10 | 150 | 0.001 |
| **Labial tentacles** | | | | | | | |
| Atrich | 37.33 ± 4.10 | 25.54–49.99 | 6.71 ± 1.28 | 3.57–11.41 | 8/10 | 868 | 0.037 |
| Microbasic b-mastigophore I | 45.36 ± 6.44 | 23.38–69.61 | 8.53 ± 1.66 | 4.32–14.76 | 10/10 | 3,180 | 0.004 |
| Microbasic b-mastigophore II | 32.74 ± 4.26 | 19.20–51.93 | 4.88 ± 0.84 | 2.69–8.31 | 8/10 | 1,620 | <0.001 |
| Microbasic b-mastigophore III | 23.47 ± 3.99 | 15.18–46.49 | 3.02 ± 0.60 | 1.46–6.80 | 10/10 | 2,369 | <0.001 |
| Microbasic b-mastigophore V | 21.50 ± 2.76 | 14.82–28.39 | 3.18 ± 0.61 | 1.72–5.08 | 4/10 | 360 | 0.004 |
| **Marginal tentacles** | | | | | | | |
| Atrich | 41.76 ± 7.40 | 25.54–62.55 | 7.64 ± 2.50 | 4.37–18.33 | 7/10 | 570 | 0.16 |
| Microbasic b-mastigophore I | 71.03 ± 8.47 | 50.32–98.38 | 12.12 ± 2.32 | 1.09–18.86 | 7/10 | 600 | 0.001 |
| Microbasic b-mastigophore II | 36.12 ± 5.66 | 20.94–58.23 | 5.42 ± 1.03 | 2.87–9.50 | 9/10 | 2,248 | <0.001 |
| Microbasic b-mastigophore III | 26.49 ± 6.15 | 15.67–48.93 | 3.54 ± 0.87 | 1.76–6.50 | 10/10 | 780 | <0.001 |
| Microbasic b-mastigophore V | 23.68 ± 4.31 | 15.08–35.75 | 3 ± 0.58 | 1.49–5.54 | 7/10 | 1,138 | <0.001 |
| Microbasic b-mastigophore VI | 53.17 ± 9.78 | 29.56–74.05 | 5.45 ± 1.36 | 1.71–9.12 | 7/10 | 687 | <0.001 |

*Actinopharynx*

The cnidom of the actinopharynx included atrichs and microbasic b-mastigophores I, II and III.

At the low level, atrichs were observed in nine out of the 10 specimens studied, while microbasic b-mastigophores I and III were quite less frequent than atrichs, being found in two and three out of the 10 specimens, respectively. Microbasic b-mastigophores II were observed in only one out of the 10 specimens.

At the middle level of the actinopharynx, microbasic b-mastigophores III were observed in six out of the 10 specimens, whereas atrichs were observed in five out of the 10

specimens studied. Microbasic b-mastigophores I were found only in two out of the 10 specimens, whereas microbasic b-mastigophores II were absent.

At the high level, the pattern was very similar to that observed at the middle level. Atrichs and microbasic b-mastigophores III were observed in five and four out of the 10 specimens, respectively, whereas microbasic b-mastigophores I appeared in four out of the 10 specimens, *i.e.*, two more than those observed at the middle level (Fig. 2A, Table S3).

*Column*

The column showed a consistent presence of atrichs at the three levels studied, being observed in 10 out of the 10 specimens studied.

The low level of the column, besides atrichs, presented ptychocysts in four out of the 10 specimens, microbasic b-mastigophores V in three out of the 10 specimens, and microbasic b-mastigophores VI in only one out of the 10 specimens. Holotrichs and microbasic b-mastigophores I were absent at this level.

At the middle level, ptychocysts were observed in six out of the 10 specimens and microbasic b-mastigophores V in only one out of the 10 specimens. Except for atrichs, no other cnidocyst types were found at the middle level of the column.

At the high level of the column, almost all cnidocyst types were recorded, except for microbasic b-mastigophores VI, which were not recorded in any individual. However, this cnidocyst type was present only in a low number of individuals. Microbasic b-mastigophores I were the most recorded after atrichs, being found in four out of the 10 specimens studied. Microbasic b-mastigophores V were recorded in only two out of the 10 specimens, whereas holotrichs and ptychocysts were found in only one out of the 10 specimens (Fig. 3A, Table S3).

*Metamesenteries*

The metamesenteries presented the least diverse cnidom of all structures, with microbasic b-mastigophores I and microbasic b-mastigophores IV. Both types were recorded in a low proportion of individuals at the three levels. Microbasic b-mastigophores I were observed in four out of the 10 specimens at each level, while microbasic b-mastigophores IV were absent at the low level and present in two and three out of the 10 specimens at the middle and high levels, respectively (Fig. 3A, Table S3).

*Labial tentacles*

In labial tentacles, both microbasic b-mastigophores I and III appeared as uniform and very frequent at the three levels. These were present in 10 out of the 10 specimens at the low and middle levels, and in nine out of the 10 specimens at the high level (Fig. 3A, Table S3). Microbasic b-mastigophores II were generally present at the middle and high levels (in seven and eight out of the 10 specimens respectively), whereas, at the low level, they were present in five out of the 10 specimens. The most particular pattern of qualitative variation in labial tentacles was that observed for atrichs, which were found in eight out of the 10 specimens at the low level, in three out of the 10 specimens at middle level, and absent at the high level. Microbasic b-mastigophores V were found in a small number of

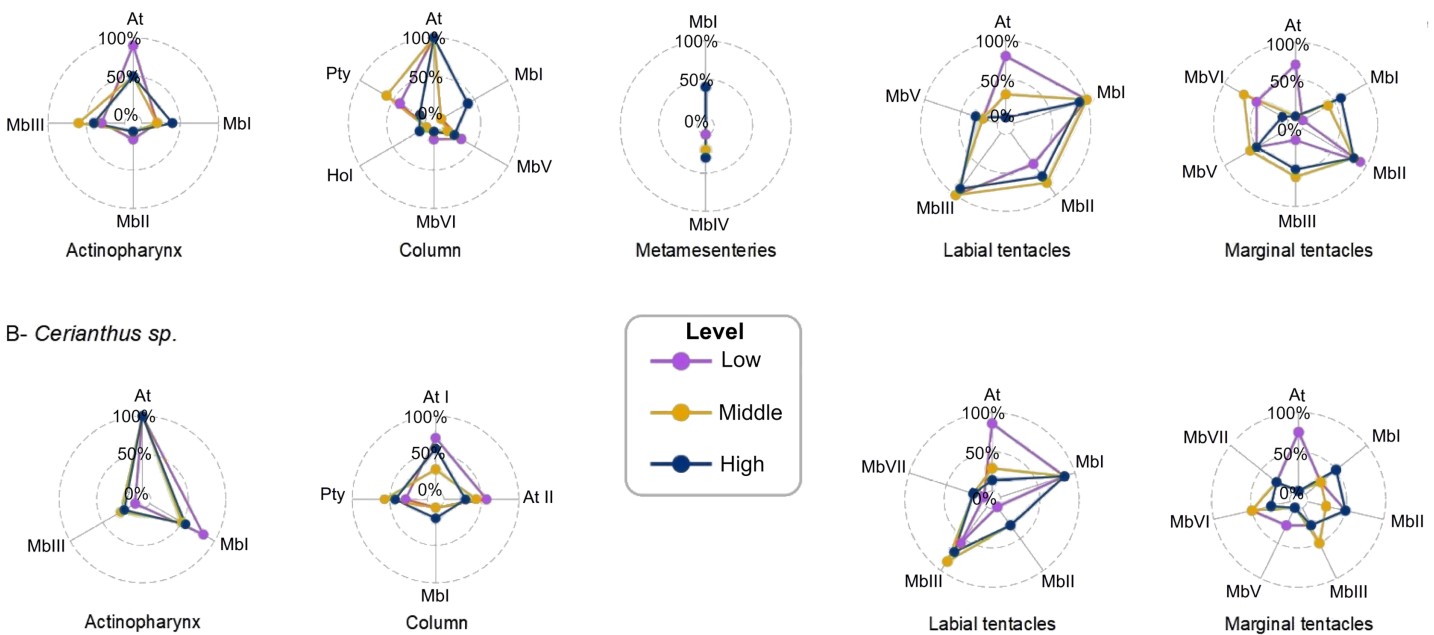

**Figure 3 Intra-structure qualitative variations of the cnidoms of *Cerantheomorphe brasiliensis* (A) and *Cerianthus* sp. (B).** At, Atrich; At I, Atrich I; At II, Atrich II; Hol, Holotrich; MbI, microbasic b-mastigophore I; MbII, microbasic b-mastigophore II; MbIII, microbasic b-mastigophore III; MbIV, microbasic b-mastigophore IV; MbV, microbasic b-mastigophore V; MbVI, microbasic b-mastigophore VI; Pty, Ptychocyst.

specimens at the three levels: in two out of the 10 specimens at the low and middle levels and in three out of the 10 specimens at the high level (Fig. 3A, Table S3).

*Marginal tentacles*

In the marginal tentacles, the cnidom pattern was similar to that observed in the labial tentacles, with several types of microbasic b-mastigophores distributed at the three levels. Also similar to that observed in the labial tentacles, a particular pattern was observed for atrichs at the different levels. However, unlike that observed in the labial tentacles, atrichs were present exclusively at the low level of marginal tentacles, in seven out of the 10 specimens, and were absent at the middle and high levels. Microbasic b-mastigophores VI were recorded in five out of the 10 specimens at the low level, in seven out of the 10 at the middle level and in one out of the 10 at the high level. Microbasic b-mastigophores II were present at the three levels in a high proportion. This cnidocyst type was found in nine out of the 10 specimens at the low level and in eight out of the 10 specimens at both the middle and high levels. Microbasic b-mastigophores V were observed with intermediate frequency at the three levels, in six out of the 10 specimens at the middle level and five out of the 10 specimens at both the low and high levels. Microbasic b-mastigophores I were not recorded at the low level, but observed in four out of the 10 specimens at the middle level and in six out of the 10 specimens at the high level. Finally, microbasic b-mastigophores III were present in only one out of the 10 specimens at the low level, and in six and five out of the 10 specimens at the middle and high levels, respectively (Fig. 3A, Table S3).

### Intra-structure variations in the cnidocyst lengths

As mentioned in "Intra-structure qualitative variations of cnidocyst types", in *C. brasiliensis*, only the atrichs of the column were observed in all specimens at the three levels sampled. Hence, both a LM and a LMM were fitted for those data sets because they adjusted to a normal distribution ($P = 0.2378$, $\alpha = 0.05$). The LMM was the best model (Table S4) and its form was: Atrich length ~ level+ (1 |Individual). The variable "Individual" was significant when comparing the LMM *vs.* the null LM (Atrich length ~ level); its standard deviation and those of the residuals of the model are shown in Table S5.

The mean estimated by the LMM showed that the sizes of atrichs from the low level were slightly larger than those from the middle level, and quite larger than those from the high level (Table S6). The confident intervals of the LMM clearly evidenced a gradient in the length of atrichs from the low to the high levels of the column of *C. brasiliensis*. The smallest sizes of the atrichs were observed at the high level of the column. Also, the CI for the length of atrichs from the high level presented the particularity that its higher size values were similar to the smallest sizes from the middle level. Moreover, the CI of the atrichs from the high level was absolutely not overlapped with that from the low level. The CIs from the middle and low levels were a little overlapped around the larger and smaller sizes, respectively (Table S6).

Comparisons between levels were also made for microbasic b-mastigophores I and III from the labial tentacles of *C. brasiliensis*. These cnidocyst types were found in almost all individuals at the three levels, with the exception of one out of the 10 specimens at the "high" level (see Table S3). For both data sets, the normality of residuals of a linear model was tested and rejected (microbasic b-mastigophores I: W = 0.99827, $P = 0.001$; microbasic b-mastigophores III: W = 0.98645, $P < 0.001$). Therefore, GLMs were fitted for both data sets of cnidocyst length. For both cnidocyst size data sets, the GLMM was the best model (Table S4) taking the following form: microbasic b-mastigophore length ~ level + (1 | Individual). The variable "individual" was significant as random effect. Its standard deviation and that of the residuals of the GLMM are shown in Table S5.

The CIs for the GLMM showed a similar pattern in both microbasic p-mastigophores I and III. A clear superposition of the size distribution of the cnidocysts was observed between the three levels of the labial tentacles for both cnidocyst types (Table S6).

The differences of sizes between levels were also explored by means of density plots (Fig. 4A). This exploration included microbasic b-mastigophores I from marginal tentacles plus the previously mentioned types in the analyses with LMM or GLMM as atrichs of the column and microbasic b-mastigophores I and III of the labial tentacles. The remaining cnidocyst types were not included in this graphic exploration because they were absent at the three levels in more than 70% of the specimens sampled (see "Intra-structure composition of the cnidom").

For the atrichs of the column, the density plots reflected the differences observed in the models, where the distribution of sizes in the levels exhibited a gradient from smallest to largest sizes from the high to the low level (Fig. 4A). Meanwhile, the density plots for

microbasic b-mastigophores, both of the labial (b I, b III) and marginal (b I) tentacles, showed a clear superposition of the distribution of sizes between levels (Fig. 4A).

### *Cerianthus* sp.

#### *General cnidom analysis*

The cnidom of *Cerianthus* sp. presented eight cnidocyst types (Fig. 5). In the column, atrichs showed two size ranges with identical morphology, and were thus differentiated as atrichs I and II. Also, spirocysts were found in the tentacles, but were not included in the analyses as they are very susceptible to mechanical variations and thus results would not be reliable. The cnidocyst types found are reported in Table 2.

The ANOVA showed significant differences in all cnidocysts analyzed whose length adjusted to normal distribution (Marginal tentacles: atrichs (F = 19.58; *P* < 0.001); b VI: (F = 17.15; *P* < 0.001); Actinopharynx: b I (F = 62.13; *P* < 0.001); Column: ptychocyst (F = 210.8; *P* < 0.001), atrichs I (F = 20.48; *P* < 0.001), atrichs II (F = 2.75; *P* = 0.029)).

GLMs were applied to evaluate differences between specimens for the remaining data sets of cnidocysts: atrichs from the actinopharynx and labial tentacles; microbasic b-mastigophores I from metamesenteries, and labial and marginal tentacles; and microbasic b-mastigophores II and III from both tentacles. In all cases, significant differences were observed between specimens (Table S7).

#### *Intra-structure qualitative variations of cnidocyst types*

Regarding the qualitative composition of the cnidom at the different levels of structures of *Cerianthus* sp., some variations were observed. The main variations were recorded in the marginal and labial tentacles, whereas in the actinopharynx, column and metamesenteries, the cnidom pattern was quite uniform between levels (Fig. 3B).

*Actinopharynx*

The cnidom of the actinopharynx of *Cerianthus* sp. was formed by atrichs and microbasic b-mastigophores I and III. In this structure, atrichs were the main cnidocyst type, being observed in six out of the six specimens studied, at the three levels. Microbasic b-mastigophores I were recorded in five out of the six specimens at the low level and in three out of the six specimens at both the middle and high levels. Microbasic b-mastigophores III were present in only one out of the six specimens both at the middle and high levels, and absent at the low level (Fig. 3B, Table S8).

*Column*

In the column, the cnidom of *Cerianthus* sp. was found to be composed mainly of two size ranges of atrichs, and ptychocysts. Both types of cnidocysts were recorded at the three levels (Fig. 3B, Table S8). Microbasic b-mastigophores I were also part of the cnidom, but appearing in a very low percentage of specimens at the high level and being absent at the low and middle levels. Atrichs I were observed mainly at the low and high levels, in five and four out of the seven specimens, respectively and in only two out of the seven specimens at the middle level. Atrichs II were recorded in four out of the seven specimens at the low
A- *Ceriantheomorphe brasiliensis*

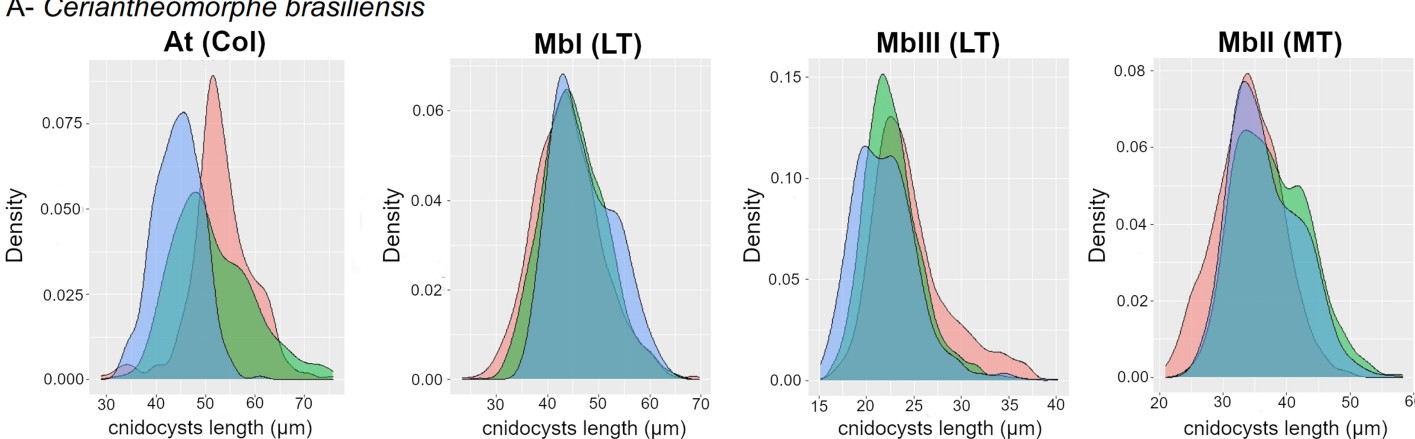

B- *Cerianthus sp.*

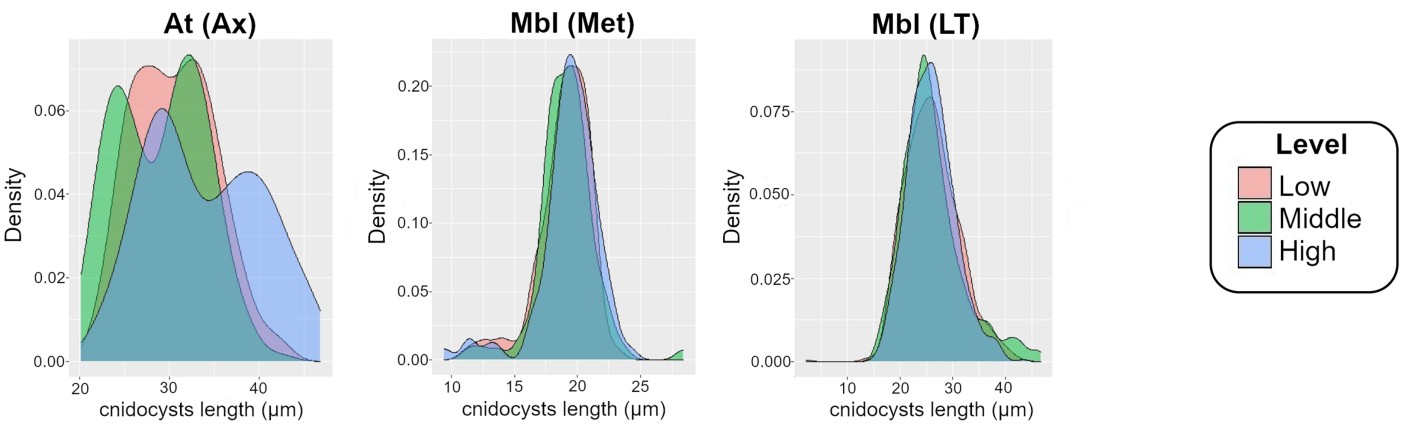

**Figure 4** **Density plot of cnidocyst lengths at the different levels of the structures of *Ceriantheomorphe brasiliensis* (A) and *Cerianthus* sp. (B).** At, Atrich, MbI, Microbasic b-mastigophore I, MbII, Microbasic b-mastigophore II, MbIII, Microbasic b-mastigophore III. Col, Column; LT, Labial tentacles; MT, Marginal tentacles; Ax, Actinopharynx; Met, Metamesenteries. Note: Only the cnidocyst types that were present in the three levels of a structure in more than 70% of the individuals studied were plotted.

level of the column, in three out of the seven specimens at the middle level and in only in two out of the seven specimens at the high level.

Regarding ptychocysts, these were observed in four out of the seven specimens at the middle level of the column, in three out of the seven specimens at the high level, and in two out of the seven specimens at the low level. Finally, microbasic b-mastigophores I were observed in the column in only one out of the seven specimens, at the high level (Fig. 3B, Table S8).

*Metamesenteries*

In the metamesenteries, microbasic b-mastigophores I were the only cnidocyst type found, being observed in all the specimens and at all the levels of the structure (Table S8, graph not included).

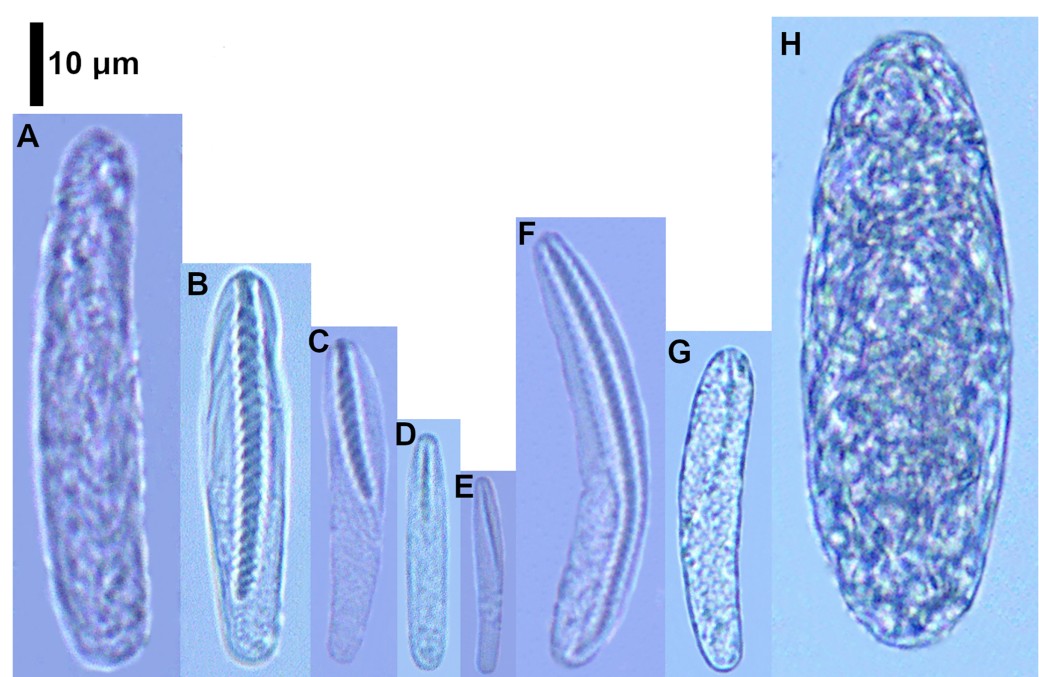

**Figure 5 Cnidocysts of *Cerianthus* sp.** (A) Atrich. (B) Microbasic b-mastigophore I. (C) Microbasic b-mastigophore II. (D) Microbasic b-mastigophore III. (E) Microbasic b-mastigophore V. (F) Microbasic b-mastigophore VI. (G) Microbasic b-mastigophore VII. (H) Ptychocyst.

*Labial tentacles*

In the labial tentacles (Fig. 3B, Table S8), microbasic b-mastigophores I appeared in a large number of specimens (six out of the seven) at the three levels. Also, microbasic b-mastigophores III were recorded at the three levels, being found in six out of the seven specimens at the middle level, in five out of the seven specimens at the high level, and in four out of the seven specimens at the low level. The other microbasic b-mastigophores types (II and VII) were found in a few specimens of labial tentacles, only at the middle and high levels (in two and one out of the seven specimens respectively), being absent at the low level. The presence of atrichs evidenced a clear variation between the levels of the labial tentacles, being observed in almost all individuals (six out of the seven) at the low level and in a small number of specimens at the middle level (two out of the seven specimens) and high level (one out of the seven specimens) (Fig. 3B, Table S8).

*Marginal tentacles*

The marginal tentacles showed several types of microbasic b-mastigophores. The distribution of this cnidocyst type was quite variable between the levels, appearing variably in one or two specimens out of the four studied at the low, middle and high levels. Atrichs were present in three out of the four specimens, exclusively at the low level, and absent at the middle and high levels. Similarly to that observed in the labial tentacles, the marginal tentacles evidenced a clear variation in the distribution of atrichs between levels (Fig. 3B, Table S8).

**Table 2 Cnidom composition of *Cerianthus* sp.** Length and width units: μm; *n* = number of specimens that present cnidocysts/total specimens; *N* = total number of cnidocysts sized. Underlined *P*-values significant at α = 0.05; # Shapiro Test was not applied due to low *N*.

| Structure/Cnidocyst type | Length (mean ± SD) | Length range (min–max) | Width (mean ± SD) | Width range (min–max) | n | N | P –value |
|---|---|---|---|---|---|---|---|
| **Actinopharynx** | | | | | | | |
| Atrich | 31.09 ± 5.67 | 20.17–46.82 | 5.65 ± 1.32 | 2.56–9.82 | 6/6 | 514 | <0.001 |
| Microbasic b-mastigophore I | 34.68 ± 5.46 | 21.41–54.93 | 6.01 ± 1.21 | 3–10.24 | 6/6 | 270 | 0.205 |
| Microbasic b-mastigophore III | 25.54 ± 5.16 | 16.52–36.90 | 2.83 ± 0.65 | 1.61–4.55 | 1/6 | 52 | # |
| **Column** | | | | | | | |
| Atrich I | 33.72 ± 3.15 | 26.46–40.80 | 9.33 ± 1.77 | 5.27–16.36 | 6/7 | 257 | 0.65 |
| Atrich II | 50.56 ± 4.24 | 40.68–62.77 | 2.86 ± 3.74 | 6.02–20.90 | 5/7 | 198 | 0.1 |
| Microbasic b-mastigophore I | 30.35 ± 2.78 | 23.26–38.91 | 6.02 ± 0.98 | 4.23–9.21 | 1/7 | 30 | # |
| Ptychocyst | 55.43 ± 10.98 | 26.33–81.38 | 21.17 ± 3.61 | 11.50–34.18 | 5/7 | 250 | 0.197 |
| **Metamesenteries** | | | | | | | |
| Microbasic b-mastigophore I | 19.12 ± 2.27 | 9.43–28.42 | 4.36 ± 0.64 | 2.59–6.81 | 3/3 | 266 | <0.001 |
| **Labial tentacles** | | | | | | | |
| Atrich | 25.29 ± 3.71 | 15.66–46.41 | 4.41 ± 0.63 | 2.67–6.88 | 6/7 | 585 | <0.001 |
| Microbasic b-mastigophore I | 26.07 ± 5.14 | 21.21–46.61 | 4.80 ± 1.08 | 3.89–9.77 | 6/7 | 1,816 | <0.001 |
| Microbasic b-mastigophore II | 24.56 ± 6.34 | 16.26–41.42 | 3.32 ± 0.76 | 1.92–5.81 | 3/7 | 127 | 0.009 |
| Microbasic b-mastigophore III | 17 ± 3.80 | 10.11–34.44 | 2.08 ± 0.48 | 1.10–4.90 | 6/7 | 982 | <0.001 |
| Microbasic b-mastigophore VII | 22.17 ± 3.44 | 16.20–32.87 | 3.47 ± 0.97 | 2.03–6.38 | 1/7 | 188 | # |
| **Marginal tentacles** | | | | | | | |
| Atrich | 26.14 ± 2.39 | 19.26–31.94 | 5 ± 0.83 | 3.54–7.54 | 3/4 | 129 | 0.581 |
| Microbasic b-mastigophore I | 32.68 ± 4.92 | 23.37–47.16 | 5.83 ± 0.96 | 3.87–8.26 | 2/4 | 298 | 0.002 |
| Microbasic b-mastigophore II | 22.67 ± 3.70 | 12.74–32.26 | 3.57 ± 0.64 | 1.79–6.05 | 3/4 | 647 | 0.014 |
| Microbasic b-mastigophore III | 19.17 ± 2.43 | 13.33–26.09 | 2.18 ± 0.24 | 1.43–2.94 | 2/4 | 294 | 0.004 |
| Microbasic b-mastigophore V | 15.14 ± 1.41 | 12.77–17.87 | 2.04 ± 0.21 | 1.60–2.46 | 1/4 | 20 | # |
| Microbasic b-mastigophore VI | 32.90 ± 5.52 | 23.83–39.06 | 4.33 ± 0.70 | 2.58–6.04 | 2/4 | 376 | 0.085 |
| Microbasic b-mastigophore VII | 22.80 ± 1.72 | 17.39–26.72 | 3.66 ± 0.38 | 2.49–5.17 | 1/4 | 365 | <0.001 |

### *Intra-structure variations in the cnidocyst lengths*

In *Cerianthus* sp., there were only two cases where a structure presented a type of cnidocyst at the three levels of all the specimens analyzed. These were the cases of the atrichs from the actinopharynx and the microbasic b-mastigophores from metamesenteries (Table S8). Neither of these two data sets of cnidocyst lengths fitted to a normal distribution (atrichs (actinopharynx): $W = 0.98527$, $P < 0.001$; microbasic b-mastigophores I (metamesenteries): $W = 0.91328$, $P < 0.001$). In consequence, GLM and GLMM were fitted and compared to obtain the best model. For atrichs, GLMM, which included the variable "individual" as random effect, was the best model (Table S9). The standard deviation of the mentioned variable is shown in Table S10. Meanwhile, for microbasic b-mastigophores, GLM was the best model (Table S9).

The CIs estimated by the model for the atrichs from the actinopharynx showed partial superposition of their length between the three levels (Table S11). The middle level

presented the lowest atrich length value (24.9 μm) according to the CI. Meanwhile, the CI for the mentioned level was almost completely overlapped with the same for the low level. The CI for the high level evidenced the highest atrich length (38.2 μm) of the actinopharynx. The CI for the cnidocyst and level mentioned was also overlapped with both the middle and low levels. However, the highest values of the CI of the atrichs calculated by the model were the exception. From the estimated average length plus 1 micron onward of the cnidocysts and levels in question, the length values of atrichs were found outside the maximum limits of the CIs of the previous levels (Table S11).

Density plots were produced both for the length of atrichs from the actinopharynx and for the length of microbasic b-mastigophores from metamesenteries of the three levels (Fig. 4B). The results of the graphical exploration of differences between levels of cnidocyst length were consistent with those of the fitted models. The density graphs for the atrichs from the actinopharynx showed overlapping curves along their length distributions between the three levels. However, the exception of the highest size values at the high level of the structure was evidenced (Fig. 4B). Besides the graph for the previous cnidocyst types analyzed above also by means of statistic models, a density plot for the microbasic b-mastigophores I from labial tentacles was produced (according to the criteria adopted, see "Intra-structure composition of the cnidom"). Microbasic b-mastigophores I from both the metamesenteries and labial tentacles evidenced clear ranges of distribution of sizes with complete overlapping between levels (Fig. 4B).

## DISCUSSION

The novelty of the present study was the new methodology used to sample cnidocysts by exploring different levels within each structure. The new methodology implemented revealed that the supposed intra-structure uniformity of the cnidom composition is not true, at least in the ceriantharian tube anemones studied. Several of the cnidocyst types observed did not present a uniform distribution between levels of structures. This was evident in the atrichs from the column of 23 cnidocyst data sets sampled in *C. brasiliensis* (considering every type present in every structure), which were present at the three levels in all the specimens analyzed. *Spier, Stampar & Prantoni (2012)* analyzed 14 data sets of cnidocyst lengths (considering every type present in every structure) and five types of cnidocysts: ptychocysts, atrichs, holotrichs, microbasic b-mastigophores I and II, whereas, in the present work, we analyzed 23 data sets of cnidocyst lengths and identified nine different types of cnidocysts. In that sense, in this research, we recorded four new types of microbasic b-mastigophores and also recorded atrichs in the labial and marginal tentacles and the actinopharynx, besides those from the column already recorded by *Spier, Stampar & Prantoni (2012)* for *C. brasiliensis*. The wider sampling at three levels in each structure used in our work could explain the new types of cnidocysts found here in relation to those reported by *Spier, Stampar & Prantoni (2012)*.

On the other hand, the atrichs from the actinopharynx of *Cerianthus* sp. were present at the three levels sampled in all specimens explored, although the number of specimens studied was low (three) and could not be a representative result.

For the remaining cnidocysts of both *C. brasiliensis* and *Cerianthus* sp., the cnidom composition presented variability between the levels in the structures. The clearest patterns of variability were observed in the cnidocysts from the labial and marginal tentacles (Figs. 3A and 3B), similarly in both species. In these structures, atrichs were present almost exclusively at the low level but not at the middle and high levels. Also, microbasic b-mastigophores I and III were recorded practically in all specimens at the three levels of labial tentacles, but were absent at the low level of marginal tentacles. A possible explanation for this pattern of atrichs could be related to the different functions of tentacles at the different levels. Penetrant b-mastigophores present at the middle and high levels of tentacles could first immobilize and capture prey. Then, volvent atrichs, linked to aggression, disposed at the level low of the tentacles and closer to the mouth, could finish the killing of the prey. The remaining structures showed no clear variability in the cnidom composition between levels. However, as observed in the tentacles, the composition of cnidocysts in the same structures of both species studied was similar. In the actinopharynx, the cnidom is dominated by atrichs at the three levels along with more variable types and less abundant endowment of penetrant b-mastigophores. This composition suggests that the actinopharynx as a whole would be the main structure linked to definitively killing prey by means of atrichs all over of it. Then, metamesenteries presented some types of penetrant b-mastigophores but in a low proportion of specimens along the three levels, suggesting that these structures would not have a preponderant function in the process of feeding or prey capture, opposite to the mentioned role for the actinopharynx. In the column, in both species studied, ptychocysts were found in a greater number of specimens at the middle level of the column. Since, in Ceriantharia, the tube can be formed in different ways according to the species, the ptychocyst may be at a specific developmental stage according to the strategy used by the animal (*Mariscal, Conklin & Bigger, 1977*; *Stampar et al., 2012*). *Stampar et al. (2012)* pointed out that the middle level of the column of *C. brasiliensis* is the zone where the tube of anemones starts its development, which is coincident with our results. In addition to ptychocysts, the column showed mainly atrichs along all of it in both species. The column also presented a more variable endowment of microbasic b-mastigophores, and holotrichs only in *C. brasiliensis*. The dominance of atrichs in the column reinforces the aggression function attributed to them, playing a defensive role against predators or other organisms.

Then, according to our findings, the classical approach of sampling to establish the cnidom composition or to compare between cnidocyst lengths, commonly used for sea anemone species, would be questionable in ceriantharians. The classical sampling proposed by *Williams (1996)*, where a particular section of each structure is clearly identified in a sampling of cnidocysts, would lose information about cnidocyst variability in all the structure. This is supported by the demonstrated qualitative variation of the cnidom between levels of a structure in the two species studied in the present work. Some authors have reported the normal distribution of cnidocyst sizes (*Williams, 1996*, *2000*; *Ardelean & Fautin, 2004*). However, other authors have found that biometry data of cnidocysts may not fit normal distribution (*Acuña et al., 2003*; *2004*; *Acuña, Excoffon & Ricci, 2007*; *Garese, Carrizo & Acuña, 2016*). Based on the results of this study, both

normality and non-normality were observed in length data of cnidocysts from ceriantharian tube anemones. These two possibilities of acceptation or rejection of normal distribution in cnidocyst length data sets are coincident with that already observed in actiniaria and corallimorpharia sea anemones (*Garese, Carrizo & Acuña, 2016*).

The lengths of the cnidocysts of both *C. brasiliensis* and *Cerianthus* sp. varied between specimens. Intraspecific variations of cnidocyst lengths have also been observed in Actiniaria sea anemones (*Williams, 1996*, *2000*; *Acuña et al., 2003*, *2004*; *Francis, 2004*; *Acuña, Excoffon & Ricci, 2007*; *Acuña & Garese, 2009*; *Acuña, Ricci & Excoffon, 2011*) and mentioned as a rule in *Garese, Carrizo & Acuña (2016)*. The present results confirm that the rule of intraspecific variations of cnidocyst sizes also occurs in the Subclass Ceriantharia.

Regarding the intraspecific variations of the lengths of cnidocysts, only the atrichs from the column of *C. brasiliensis* evidenced some differences in the length of cnidocysts between levels. This cnidocyst type showed a gradient of size variation. The highest length of cnidocyst values were observed at the low level and decreased at the middle and high levels (Table S6). The atrichs of the column from the low level were 5.38% larger than those from the middle level and 17.56% larger than those from the high level. *Ardelean & Fautin (2004)* studied the length of microbasic b-mastigophores from the column of one specimen of *Actinodendron arboreum* (Quoy & Gaimard, 1833) and found that the mentioned cnidocyst type presented less standard deviation and a narrow range of lengths of cnidocysts at the middle level of the structure than at the low and high levels. Similarly to that observed for the atrichs from the column of *C. brasiliensis*, the microbasic b-mastigophores from the column of *A. arboreum* presented the lowest lengths of cnidocysts at the high level and the longest ones at the level low of the structure. *Robson (1988)* suggested that the variation of cnidocyst sizes may be the result of the different stages of development of cnidocysts, and that the high variability in the sizes and types of cnidocysts between individuals of the same species can be explained by the interaction between the demand and replacement of the product of intracellular secretion. Then, a possible explanation of the intra-structure gradient observed in the column of *C. brasiliensis* could be attributed to the burrowing form of life of ceriantharian sea anemones, which would make the high level of the column more exposed and then the use and replacement of the cnidom could be more frequent, inducing the presence of cnidocysts that are not completely developed and hence have smaller sizes. Finally, atrichs were the most abundant type of cnidocyst in the column in both species, more than ptychocysts (which are exclusive of Ceriantharia).

## CONCLUSIONS

This is the first study carried out on the variation of the composition and size of cnidocysts in the Subclass Ceriantharia, with a considerable sample number. Based on the results, we can conclude that the size of cnidocysts in ceriantharian sea anemones vary intraspecifically, a fact that is a rule in other groups (*Acuña et al., 2003*, *2004*; *Francis, 2004*; *Acuña, Excoffon & Ricci, 2007*; *Acuña & Garese, 2009*; *Acuña, Ricci & Excoffon, 2011*; *Garese, Carrizo & Acuña, 2016*). The data obtained in this study reinforce the observation

of authors such as *Schmidt (1972)* and *Fautin (2009)*, who reported that the variation of the cnidom between individuals of the same species is sometimes higher than that between individuals of different species. Our results also prove that tube anemone species could present both qualitative variations of the cnidom and intra-structure variations of the cnidocyst sizes. The intra-structure cnidocyst variation could imply different functions at different levels of a particular structure of the organism. The new findings presented open new questions for further research, such as how these variations could be showing different functions of the different levels within a specific structure of the anemone body or whether these variations are an exception of ceriantharian tube anemones or could be found in sea anemones such as the actiniaria ones. If the last scenario was verified, it could call in question all the previous descriptions of the cnidom of sea anemone species.

## ACKNOWLEDGEMENTS

We thank Dra. Florencia P. Bamonte for editing the figures, Lic. Nicolás Vazquez for his suggestion in graphic presentations, and Fabrizio Scarabino for providing specimens of *Cerianthus* sp. from Uruguay. Also we thank to The International Association for Biological Oceanography (IABO) to enable the publication of this MS. We really appreciate and thank the constructive comments and revisions by Igor Kosevich, as well as the comments of the anonymous reviewer, which have greatly contributed to improve this work.

### Funding

This work was supported by grants of Agencia Nacional de Promoción de la Investigación, el Desarrollo Tecnológico y la Innovación (PICT 2016–Serie A, N°0879), Universidad Nacional de Mar del Plata (UNMdP), Argentina (EXA1037/21), PIBAA CONICET (28720210100435CO) to Agustín Garese. Sergio Stampar was supported by São Paulo Research Foundation (FAPESP), grant number 2019/03552-0, and CNPq (Research Productivity Scholarship) grant number 301293/2019-8. Fabiola Corrêa was supported by FAPESP (N°2018/21566-6). The funders had no role in study design, data collection and analysis, decision to publish, or preparation of the manuscript.

### Grant Disclosures

The following grant information was disclosed by the authors:
Agencia Nacional de Promoción de la Investigación, el Desarrollo Tecnológico y la Innovación: PICT 2016–Serie A, N°0879.
Universidad Nacional de Mar del Plata (UNMdP) Argentina: EXA1037/21.
PIBAA CONICET: 28720210100435CO.
São Paulo Research Foundation (FAPESP): 2019/03552-0.
CNPq (Research Productivity Scholarship): 301293/2019-8.
FAPESP: N°2018/21566-6.

## Competing Interests

The authors declare that they have no competing interests.

## Author Contributions

- Agustín Garese conceived and designed the experiments, performed the experiments, analyzed the data, prepared figures and/or tables, authored or reviewed drafts of the article, and approved the final draft.
- Fabiola Goes Correa conceived and designed the experiments, performed the experiments, analyzed the data, prepared figures and/or tables, and approved the final draft.
- Fabián H. Acuña conceived and designed the experiments, authored or reviewed drafts of the article, and approved the final draft.
- Sérgio Nascimento Stampar conceived and designed the experiments, authored or reviewed drafts of the article, provided the specimens studied, and approved the final draft.

## Data Availability

The raw data is available in the Supplemental Files.

## Supplemental Information

Supplemental information for this article can be found online at http://dx.doi.org/10.7717/peerj.15549#supplemental-information.

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
