# Peer review of "Cnidom in Ceriantharia (Cnidaria, Anthozoa): new findings in the composition and micrometric variations of cnidocysts"

_PeerJ, doi:10.7717/peerj.15549_

## Round 0.1 · original submission · Major Revisions

Dear Dr. Garese and coauthors,

I am agreeing with both reviewers that the manuscript needs serious English language improvement, corrections of the tables and figures legends, some introduction/discussion improvement. Also, reviewers noted that your manuscript includes many problems with terminology and wording.

Best regards,
Alexander Ereskovsky

·

Basic reporting

1. Even as a non-native English speaking person I feel that the manuscript is written in very poor English. That causes real difficulties in understanding authors’ narrations and statements. Moreover, it looks like the manuscript is written negligently and contains many misprints and incorrect usage of the terms (e.g. pticocyst instead of ptychocyst; the same structure in different parts of the manuscript is called mesenterial filaments, then mesenteries, or metamesenteries; intra-structure – intraestructure; etc. (they are marked with a colour in the annotated manuscript)). Often the style of the manuscript resembles slang (sentence fragments) (e.g. line 119: “The normality of the capsule length was tested by the Shapiro-Wilk test…”).
The English language should be improved to ensure that an international audience can clearly understand the text.
2. The manuscript includes sufficient introduction and background to demonstrate that the work fits into the broader field of your knowledge on cnidarian cnidom. At the end of introduction, I would suggest to strengthen the ‘novelty’ of this research with more clear explanation what means ‘the intra-structure approach’. Relevant prior literature referenced appropriately.
3. The manuscript conforms to acceptable format.
4. Figures and tables included into the manuscript are relevant to the content. However their descriptions and labelling needs improvement and corrections (pointed in the annotated manuscript).
5. Reference list contains some extra references, not mentioned in the text. In addition, it looks like there are some mistakes in the order of the Acuña F.H. references.
6. Supplementary material contains 10 tables including raw data on cnidocysts measurements and statistical criteria. Including the Raw Data is appropriate, but the file needs clear description of all abbreviations used, and replacement of the Spanish terms by English ones. The rest remarks and suggestions see in the annotated files.
7. As for the figures and tables in the manuscript and supplementary material – all abbreviations must be uniform everywhere – otherwise it confuses the reader.
8. On a whole, the submission is more or less ‘self-contained’ (if the mentioned problems are not considered).

Experimental design

1. For research conducted on non-regulated animals, a statement should be made as to why ethical approval was not required.
2. The submission deals with the primary research within the aims and scope of the Journal.
3. The submission clearly defines the research question, which is relevant and meaningful. The knowledge gap being investigated identified, and necessary statements how the study contributes to filling that gap are made at the end of introduction. I propose to strengthen them more.
4. The investigation had been conducted rigorously and to with a high standards of data analysis.
5. Methods used to analyse the data are applied correctly and are described with sufficient information to be reproducible by another investigator (some minor comments are added in annotated files). The only disputable point is application of GLMM model for comparison of mean (average) cnidae length (absolute (estimate) values) instead of classical non-parametrical statistical methods. Foe analysis of the trends GLMM model fits exactly.
6. The research had been conducted in conformity with the prevailing ethical standards in the field.

Validity of the findings

1. The data are robust, statistically sound, and controlled. The raw material provided in supplementary files. All underlying data have been provided; they are robust, statistically sound, & controlled.
2. The conclusions are appropriately stated, connected to the original question investigated, and are limited to those supported by the results. However, the discussion in general could be improved on the bases of the functional differences of cnidae types and differences of specimens (size and locality). That will widens the audience of the research.

Additional comments

The manuscript needs serious English language improvement, corrections of the tables and figures legends, some introduction/discussion improvement.

Reviewer 2 ·

Basic reporting

The MS presented for the review is rather accurate study of the tube anemone cnidae for two species that was accomplished to evaluate intraspecific and intra-structure variations in cnidom composition. MS follows the IMRaD structure. Figures of a good quality and raw data is shared. Literature coverage is quite appropriate. Apparently I would advice to include Ostman 2000 for general overview, but it is up to authors. There are some problems with numbering of parts of the MS . Thus, Material and Methods and Results sections have the same numbering (2.X)
There are some problems with terminology and wording. E.g the authors use wording “proximal” and “distant” to denote studied parts of the animals. The terms proximal and distant are not appropriate to use withing ceriantharian with anything but tentacles, as these two words are used to describe parts of a feature that are close to or distant from the main mass of the body. It is absolutely unclear for me what authors can describe as distal part of the column (see discussion section). The wording “segment” used in the MS seems inappropriate for non-segmented animal. Also, please, do not use “sea anemone” do denote ceriantharians. "Ceriantharians" alone or "tube anemones" will do. The MS would profit from some condensing because of wording used and I encourage authors to use shorter and less complicated sentences.

Experimental design

Formally the MS is an original research within Aims and Scope of the PeerJ. Authors tried well in introduction to define research questions and the knowledge gaps, however, the MS would profit from some condensing because of wording used and I encourage authors to use shorter and less complicated sentences.

I am sorry to inform authors that Material and Methods is far from accuracy and needs a throughout revision. From the raw data it is not clear even how many specimens of Cerianthus sp were used (there are numbers 1-5, and also 5.1, 5.2 and 11 that makes 8, but not 7 as reported at line 104 of the MS

1. For each specimen collected authors have to provide accurate data (like specimen number (ideally museum numbers, however, unique Acc or collection number used by author will do), coordinates and depth of the collection, date of collection, name of collector, name of identifier, and probably any other additional data that may influence the cnidom like type of substratum, associated fauna, density of animals, other anthozoans/cnidarians nearby, predators or traces of predation, size etc). It is really important to have this information taking into account that material came from several rather distant locations. Neither section 2.1 (lines 104-106) nor Supplementary Table S1 provide this information.
2. I kindly request authors to re-format the raw data so it can be compared easier. All ten specimens of Ceriantheomorphe brasiliensis and 7 (?) specimens of Cerianthus sp have to be placed as adjacent columns – so it would be possible to compare the specimens, e.g. for presence/absence of particular type of cnidae. For each specimen in raw data file please provide unique number from Supplementary Table S1 (see comment 1 above) so reader can see if specimens from different locations demonstrate any difference in cnidome.
3. Section 2.2. lines 108-112.Need some wording to fix to be more reader friendly. So authors have to begin with (1) what body regions were sampled for cnidae (marginal tentacles, Labial tentacles, column, actinopharynx, mesenterial filament); (2) how authors chose tentacles (outer tentacles ? Inner tentacles? Tentacles from the same quartet?) (3) how authors decided to subdivide structures as “base”, “middle” and “tip” for each region sampled – in this case we probably need an additional illustration showing subsample protocol. What was the sample size? It is not clear how author chose between base, middle and tip in case of actinopharynx (also if siphonoglyph and/or multiplication zone were samples (if this info was neglected if it may affect cnidae distribution between specimens)), column and mesenteries. (5) In case of mesenteries. As I see from the raw data file, authors used metamesenteries for cnidae studies, however it is not indicated what regions of mesenterial filament (ciliated tract, cnido-glandular tract, craspedia) or which metamesenteries (M-,B-,m- or b-) were used in analysis. In case if authors used for measurements different regions of mesenterial filament in different specimens it may affect seriously the results as cnidoglandular tract and ciliated tract have different cnidom. Without explanation of terminology in M&M section, trying to get through Results and Discussion , the reader cannot get any idea where the discussed type of cnidae came from and what authors are writing about. Illustration of subdivisions is critically needed.
4. It is absolutely necessary to describe how the preparations for cnidae studies were made. There is no a single word about this (even if it is a routine practice it has to be explained here). There is no info what optics (microscope? DIC?) and camera authors used.
5. It is not clear for me why authors measured only nematocysts and ptychocysts, but left spirocysts out. It has to be either reasonably explained or spirocysts have to be included.

Without clearing up these above-mentioned points 1-5 we cannot even assess validity of the statements and speculations made by authors.
Few random comments to the M&M section of the MS
line 114 Have in mind that ptychocysts were described in 1977 so authors cannot use terminology by Mariscal 1974 (see line 114) only, please make necessary changes.
line 116 . “The cnidom was described and their sizes compared ..” Cnidome is composition of cnidae and singular you cannot use “their size” - please rephrase
Lines 161-168. Figure 1. Table 1. Authors need to state somewhere the differences between cnidae types discerned during the study (same for another species). It is unclear for me the differences between e.g. b-mastigophores III and V. What additional categories were found apart of those reported Spier, Stampar & Prantoni (2012) (line 163).

Validity of the findings

Definitely such throughout analysis is novel. However, there is a tendency that authors overvalue their results as " the first study carried out on the variation of the composition and size of cnidocysts in Ceriantharia". It is mainly complicated statistical analysis that was used by authors.
Unfortunately there are some serious questions for underlying data.

I have very serious concerns about authors findings due to misinterpretation.
The term “atrichs” was broadly used in the old literature for ceriantharians (see e.g. Carlgren 1940, Schmidt 1972, 1974, den Hartog 1977). However, in 1977 Mariscal describes a new category of cnidae having principally different way of folding of undischarged thread in the capsule. He called this new category “ptychocysts” (from Greek “ptychos” meaning “fold” please, check spelling for Ptychocysts and other cnidae throughout the MS). I have very hard concern that “atrichs” reported and discussed by authors of the MS are t ptychocysts in slightly different projection (see figure 25 a-c in Mariscal, 1977 cited by the authors and compare a-b and c). For me from Figures provided by authors (Fig 1A and Fig 4A) they are obvious ptychocysts. However, if authors would like to insist on atrichs, they need to provide TEM sections showing threads of these capsules to be helically folded but not pleated in undischarged capsule. Otherwise atrichs have to be combined with ptychocysts and analyzed accordingly and all necessary changes in results, discussion, conclusions, tables and supplementary tables, and illustrations has to be made. Apparently this change will change dramatically results and conclusions of the MS.

Additional comments

not needed

---

## Round 0.2 · Minor Revisions

Dear Dr. Garese,

You and your coauthors have addressed all of the reviewers' comments. I am happy with the current version. Unfortunately, the abstract and manuscript are still a bit "clunky" in the sense that they are understandable but the wording is a bit odd. e.g. lines 27, 37, 39-40, and many more in the abstract. I strongly advise you to have the manuscript read and corrected by a person who speaks fluent English, or a language service. Unfortunately, without these corrections, this paper cannot be published.

Best regards,
Alexander

---

## Round 0.3 · accepted · Accept

Dear Dr. Garese, thank you for your work in improving the English in the manuscript. The manuscript is now ready for publication.
Best regards,
Alexander